# Satisfaction with Selected Indicators of the Quality of Urban Space by Polonia in the Greater Toronto Area

**Kamila Ziółkowska-Weiss**

Department of Tourism and Regional Studies, Institute of Geography, Pedagogical University of Cracow, 30-084 Kraków, Poland; kamilazw@up.krakow.pl

**Abstract:** The main objective of this article is to determine the quality of life of Polonia living in the Greater Toronto Area (GTA), with particular emphasis on urban quality, which influences their assessment of the standard of living in this city. The presented results of the research are based on a survey questionnaire conducted with the participation of 583 respondents. The respondents evaluated, among others: accessibility to recreational tourism in the city, public transport, possibilities of finding a job, accessibility to housing and quality of the natural environment. Assessment of the selected indicators was correlated with the application of the statistical coefficient of the chi-squared test with particular sociodemographic characteristics of the examined respondents (with age, place of residence of the respondents (Toronto, suburbs) and their duration of residence in the GTA). On the basis of the formulated research hypotheses and conducted studies, it can be concluded—among others—that the satisfaction level with regard to accessibility to housing increases with age, that people living in the GTA suburbs rate accessibility to transportation lower than people living in Toronto and that people living in the GTA for more than 20 years rate accessibility to tourism, leisure and relaxation lower than people living in the GTA for a period shorter than 20 years.

**Keywords:** quality of life; place space; Polonia; Toronto; Canada

## 1. Introduction

The city is an urban space where the exchange of products, goods, services, information, sensations and experiences between different groups of users of the same space has been taking place for centuries. Thus, cities can be regarded as spaces of exchange, used in various ways. The functioning of the city, its spatial and social structure and governance process belong to a group of issues that are frequently addressed in works published in different disciplines: urban geography, urban planning, spatial planning, history, economics, urban policy, sociology and cultural studies [1–8].

Among the problems discussed by researchers, we find issues related to various aspects of urban living conditions: architectural form and urban layout, morphology, social composition, functional equipment and efficiency of the city as an economic unit, quality of the urban landscape, living conditions of the residents and their access to culture, access to tourism and recreation in the city and image and perception of the city [9–20].

Urban space itself is considered to be a "distinctive fragment of geographical space" [21], which "is formed by people, social groups, institutions, activity systems, etc." [2] It is created as a result of the use of the city-forming factors, positively translating into the targets of the individuals seeking their place on Earth. Urban space is a condition for existence of the urban environment [7], which determines the behaviour of all groups of city users, so that they mutually influence each other when functioning within it. The reference to space and place in which quality of life is analysed is presented by a definition of the World Health Organization (WHO) of 1992, stating: "an individual's

perception of his or her position in life, in the context of the culture and value systems accepted by the society in which he or she lives, and in relation to his or her life goals, expectations, interests". In view of this, quality of life constitutes "a set of spatial-environmental, production and cultural factors that comprise the reality in which a person lives" [19]. According to Zadrożniak [22], the factors shaping the quality of life of the inhabitants in the basic dimensions of the city development concern economic aspects (development of road infrastructure, access to tourism and recreation in the city), social aspects (accessibility to education, situation on the labour market, situation on the real estate market), environmental aspects (system of nature protection in the city, emission standards of environmental pollution) and institutional aspects (access to culture, care for cultural heritage in the city). These aspects have been taken into account in the presented article by the author.

While analysing the subject literature, publications concerning the broadly defined level and quality of life in cities are available [23–31], with visions of the development of the living conditions in the cities [32–37] or quality of urban life versus the role of transport [38–45].

This article pays special attention to the issues related to urban space and assessment of the quality of life in the GTA by Polonia, who constitute a large part of the Canadian community. This is a new issue that has not been addressed in the literature before.

The history of urban development in Canada is divided chronologically into several stages (their periods are given in brackets): colonial (1608-1800), early industrial (first half of the 19th century), industrial expansion (from the 1870s to the outbreak of World War II), wartime and post-war "boom" (from the 1940s to the end of the 1960s), decentralisation and suburban development (from the 1970s to the 1980s) and secondary decentralisation and diversification of urban regions (turn of the late 20th and early 21st centuries) [46,47]. Each of the distinguished stages characterising the urbanisation process in Canada is reflected in the urban layout, functional structure, economic development, social and cultural relations, image and perception. The above-mentioned issues are discussed in the publications devoted to broadly defined studies of the Canadian cities, including Toronto, where the research was conducted [48–53]. In 2001, the United Nations awarded Toronto the title of the World's Most Multicultural City [54]. Today, more than half of the world's people live in the cities. In this situation, the need for designing a friendly, sustainable and usable space is becoming greater than ever before.

It is worth analysing whether the quality of urban space is rated as highly by inhabitants of the agglomeration as by tourists coming to the city. Considering the fact that there are 259,715 people of Polish origin in the GTA, which constitutes 4.04% of the total population of the area, assessment of the quality and standard of living by this national minority living among the representatives of more than 200 ethnic groups [55] is significant and constitutes an important voice in the overall assessment of urban space by all residents of the GTA.

There are numerous publications of Polish specialists in Canadian studies, who address the following aspects in their studies: history [56–68], migration [69–71], Polish pastoral activity [72,73] and the causes of emigration and Canada's policy towards receiving emigrants [74,75].

In the vast literature describing the subject of Polonia in Canada, as well as the quality of life in the city, there has not yet been a scientific publication analysing the assessment of urban space among the Polish community living in the Greater Toronto Area, taking into account selected factors and correlated with the sociological and demographic characteristics of the respondents. The results of the research [76,77] on selected issues of the level and quality of life by Polonia in the GTA are available, yet they do not include the statistical analysis and verification of the research hypotheses. The previously presented results of the research provide a constructed general model that can be applied to assess the quality and level of life of the selected social groups around the

world. Therefore, the presented article constitutes an important issue from a geographical, sociological, migration and cultural point of view, complementing the information on the assessment of the quality of life of the Canadian Polish community in the GTA. The statistical data confirm that, despite the relatively restrictive immigration regulations and plans for their additional tightening, Canada still remains an attractive country for newcomers. The quality of life in the city and evaluation of the attractiveness of the particular aspects for the overall satisfaction of its inhabitants is important from the point of view of local government policy as well as administrative investment and development plans.

In the presented article, the author has put forth several research hypotheses, focusing her attention on the selected aspects of satisfying the needs of living in an agglomeration, correlating them with the demographic and social characteristics of the studied group of respondents. The posed research hypotheses assume the following: satisfaction ratings of accessibility to housing increase with age, people over 40 assess the opportunities of finding a job in the GTA very well, people living in the GTA suburbs rate accessibility to transportation worse than people living in Toronto, people living in the GTA suburbs rate the natural environment more highly than people living in Toronto itself, and people living in the GTA for more than 20 years rate accessibility to tourism, leisure and relaxation lower than people living in the GTA for the period shorter than 20 years. The hypotheses have been posed in such a way so that it is possible to determine the level of life satisfaction by the Polish community in the urban space of the GTA with their application, which constitutes the main objective of the article.

## 2. Materials and Methods

### 2.1. Spatial Scope of the Empirical Research

The contemporary distribution of Poles in Canada was conditioned, on the one hand, by the processes of development of the young state and, on the other hand, by social changes in the Polish community itself. Until the 1930s, immigrants from Central Europe were directed to agricultural areas in the provinces of Manitoba, Saskatchewan and Alberta by Canadian authorities. It was only after the Second World War that Poles began to settle mainly in cities. Urbanisation processes resulted in the fact that, as early as in the 1970s, 80% of Poles lived in cities, mainly in the large agglomerations of the provinces of Ontario and Quebec. Nowadays, cities, particularly Toronto and subsequently Montreal, Ottawa, Edmonton, Hamilton, Vancouver, Winnipeg and Calgary, constitute the main centres of the Polish community. Representatives of the most recent waves of the Polish migrants from the second half of the 20th century settled mainly in Toronto, hence the strong numerical dominance of the Polish community living there. The most active organisations operate in this city and most of the Polish press titles are published here [59].

Canada is a country of immigrant origin, which attracts people from all parts of the world due to the offered standard of living. Toronto constitutes a modern multicultural metropolis that embraces the entire spectrum of social and cultural diversity. It is a manifold space—taking into account the ethnic and racial diversity of the residents—that respects the symbols attributed to it. Multiculturalism, equally as strong, manifests itself—above all—in the ethnic diversity of the inhabitants of the contemporary cities. The population living in contemporary Canada is predominantly of immigrant origin. Currently, the indigenous inhabitants of these areas constitute only approximately 4.9 % of the total population of the country. Therefore, Canadians are newcomers, often in the first generation. According to data for the period 2000-2019, around 390,000 migrants from all over the world settled here each year.

In total, 35,151,728 people lived in Canada in 2016, where 1,106,585 inhabitants were of Polish origin. This is 3.9% of the total population. Ontario, where the Greater Toronto Area is located, constitutes the province inhabited by the largest number of Po-

lonia. Overall, 523,490 people lived in Ontario in the year 2016, while the largest number of people of Polish descent in relation to all residents is in Manitoba, where the percentage of Poles is as high as 6.9%. The provinces of Nunavut and the North-Western Territories are inhabited by the smallest number of people of Polish descent.

The Greater Toronto Area includes the regions of Peel, Halton, York, Durham and the city of Toronto itself. In total, 259,715 people of Polish origin live in the Greater Toronto Area, which is 4% of the total population in this area. Most Polish people live in Mississauga, in the county of Peel. This population equals 43,350 people, which is 6% of the total population in Mississauga. A significant number of people of Polish origin live in the county of York, in Vaughan. This number includes 18,265 people, which is 5.96% of the total population in the city. The smallest number of Polish people live in two towns in the county of Durham. This is the city of Brock, with 335 people of Polish descent, and the city of Scugog, with 590 people.

Figure 1 shows the percentage of Polish people in the Greater Toronto Area in relation to the total population of the entire area in 2016.

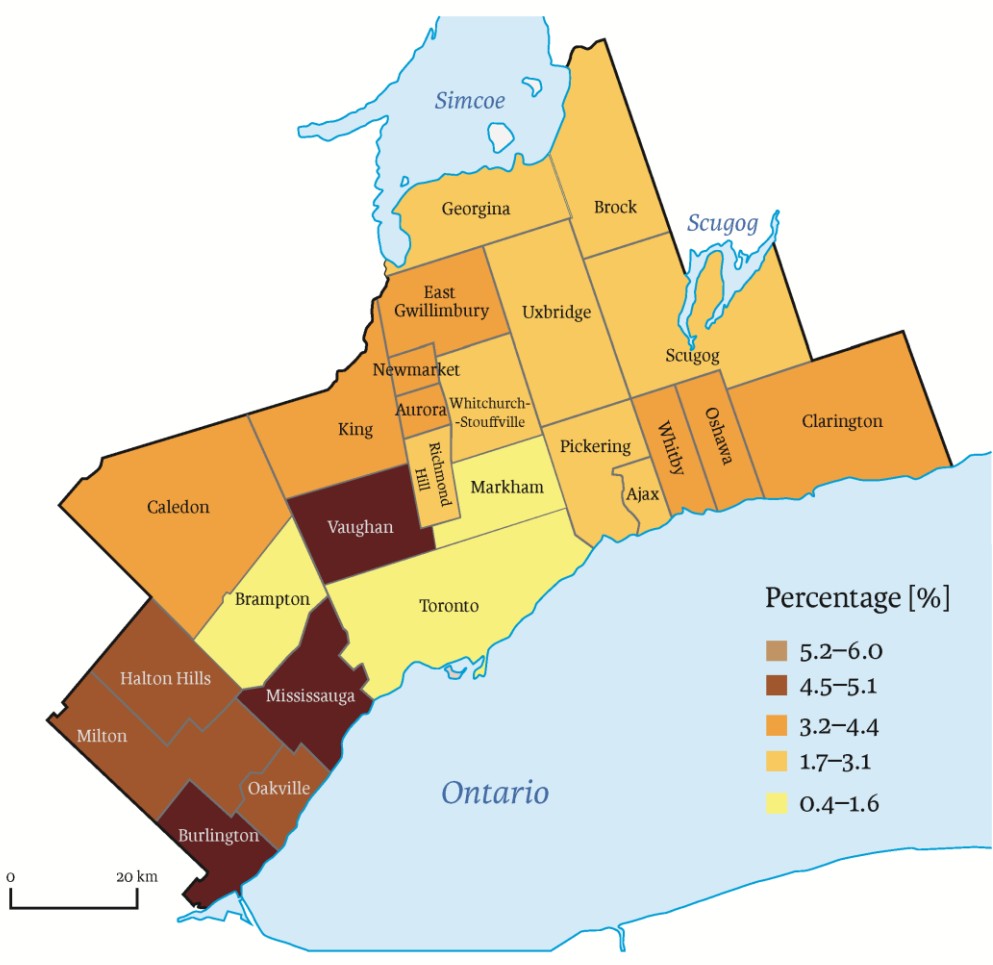

**Figure 1.** Percentage of Polish people in the Greater Toronto Area in relation to the total population of the area. (Source: Author's own compilation based on the conducted research.)

As more immigrants came from Poland, and as a result of the functioning of migration social networks, significant concentrations of Poles were formed in Toronto. Consequently, "Polish blocks of flats" were created, e.g., in the vicinity of Lakeshore Street at Lake Ontario, but also in the blocks of flats in Scarborough and other districts of the city

and in Mississauga adjacent to Toronto. In the latter, a large concentration of Poles lived in the vicinity of the Maximilian Kolbe Polish Church. The Polish district along Roncesvalles Street in Toronto also constituted a place where many Poles were concentrated.

The main area of the research was the Greater Toronto Area, which was selected by the author due to the presence of the largest Polish community in Canada.

Toronto is a typical American city, with an urban grid plan, which extends over an area of 630.2 square kilometres. The origin of this city conditioning its contemporary spatial composition also influences the functional structure. The regular arrangement of the rectangular streets and corresponding quarters of a similar size means that these are mostly urban buildings, their architectural form and their functional purpose constituting the factor distinguishing specific fragments of the city.

### 2.2. The Time Period of the Empirical Research and the Research Tool

The time period of the conducted research covered the years of 2017, 2018 and 2019. In 2017, the pilot studies were conducted, whereas the actual research in Canada was carried out in two stages. The first stage of the field studies took place during the period from 16 August 2018 to 18 September 2018, while the second stage from 10 September 2019 to 19 September 2019.

During the implemented research, the author applied quantitative methods. The quantitative methods are commonly associated with studies where the numbers are used to characterise the properties of observation units and/or to present the research results (although, in fact, other, "non-numerical" formal objects, e.g., graphs or relationships, are applied). Quantitative research is identified with statistical methods, which is justified as they occupy a dominant position among the quantitative methods. Providing the explanations by measuring the examined phenomena constitutes the essence of quantitative research. The results of these kinds of research are subjected to mathematical and statistical analyses that enable determination of the causes of their occurrence as well as the connections and relationships that occur between them, which—in turn—is used to determine the principles and regularities and sometimes laws relating to the examined reality. In quantitative research, the fundamental activity is measurement, without which it would lose its quantitative character. Most often, this measurement refers to the frequency of occurrence of the examined phenomena or the degree of severity of the examined feature [78].

One of the quantitative studies is a diagnostic survey method in the form of questionnaire surveys. This method has been applied in this research. The results of the research conducted among the Polish community living in the Greater Toronto Area constituted the fundamental empirical base. The empirical part of the research took the form of the diagnostic survey, conducted with the application of a representative method among the population of Polonia of the Greater Toronto Area. The appropriate research covered a total of 612 respondents, including 583 questionnaires used for the analysis.

The survey questionnaire was prepared in two languages, Polish and English, and consisted of 17 questions and 15 questions in the personal information section. In quantitative research, answers are most often "provided" and respondents indicate one of them.

To ensure that the sample group was representative, Census Canadian data from 2016 and 2019 were used to calculate the gender and age structure of individuals in each age group relative to the entire Greater Toronto Area Polish population. The most recent census, which includes people of Polish ancestry and is available, is from 2016. It was first used to calculate that there are 523,490 people of Polish ancestry living in Ontario, of which 251,575 are men and 271,910 are women. The next step was to calculate how many people of Polish ancestry live in the studied area. There are 113,300 men and 123,940 women living in the GTA, for a total of 237,245 people. The methodology of the research assumed that the author would examine 0.25% of the population of Polish origin living in the region. After calculations, this gave a total of 593 people, including

283 men and 310 women. For the research to be representative and reliable, it was necessary to calculate the age and gender structure, according to the methodological assumptions. Publicly available censuses do not take into account the current age and gender breakdowns of representatives of the various national minorities in Canada. Therefore, it was assumed that, based on publicly available age and gender breakdown data, the author would perform calculations for Ontario residents to determine what percentage of a particular age group is male and what percentage is female. The survey was conducted by age and gender, and 583 properly completed survey questionnaires were used for the final analysis. After establishing the age and gender structure of the Ontario population, the author determined, by calculation, the quantitative structure of the Polish community residing in the GTA.

In order to make the research as representative as possible, seven age groups were differentiated, with the use of the pre-conceived division. These were the following age groups: 15-19, 20-24, 25-29, 30-39, 40-49, 50-59 and over 60 (where women and men were additionally included separately). Conducting research among people aged 15 and over (youth) allowed certainty that the respondents understood the asked questions and answers provided by them were reliable. In this article, the research results are presented for all age groups together.

However, the purpose of this article was not to indicate the quality of living standards in particular age groups, but to capture the statistical relationship in the surveyed population of Polish diaspora, and to indicate general characteristics for this ethnic group in multicultural Toronto.

As indicated by the results of the study, the Polish community living in the Greater Toronto Area is characterised by either a high school or college education. The highest percentage of respondents with college education was found in the age range of 30-39 years, where they accounted for 14%, and in the age group over 60 years—14.7%. In the most numerous group of respondents, over 60 years, 12.6% of respondents have a university degree, and 11% of respondents have high school education. Regarding the Polish community living in the Greater Toronto Area, their overall financial situation is assessed as good by 53.3% of respondents, and as very good by 23.2%. It was considered average by 16.5%, and bad by 5.0%. Only three persons in the age group of 15-19 assessed their material situation as very bad; there were no such responses in other age groups. As many as 42.2% in the age group of 30-39 years rated their material assessment as very good, and in the age group of 50-59 years, there were 29.1% such people.

### 2.3. Statistical Methods Applied in the Conducted Research

The chi-squared test, for computer-processed statistical material (constituting a specific database) subjected to statistical analysis based on the statistical methods, enabling the detection of regularities in the examined community, constituted the main statistical method applied in this work. The following methods were selected and used at this stage:

- the mean and the median—for analysis of the community structure;
- non-parametric chi-squared test of independence—applied to determine the significance of relationships between variables. The choice of the test was conditioned by the following facts:
- the sample size was large as compared to the size of the general population—a necessary condition for the application of the non-parametric tests of significance [79];
- measurements were made at the level of the nominal and ordinal scales [80].

The chi-squared test of independence, which allows us to determine statistical significance for a relationship between two variables measured on a nominal scale (i.e., qualitative data), was applied in order to identify differences in particular research groups and in an attempt to answer the hypotheses posed at the beginning of the elaboration. The chi-squared test of independence (chi2 ($x2$)) is based on comparing the num-

bers of responses obtained in the surveys with the numbers that can be expected, assuming that there is no relationship between the analysed variables. The basic values calculated in this test are chi2, the degree of freedom (df) and the level of statistical significance (alpha). The "alpha" value that determines the probability of obtaining a specific effect in the sample is of elementary importance for inference if this effect was not present in the population [80]. This indicator is used to assess whether the obtained result is statistically significant. In the presented elaboration, as in most studies in this area, it has been assumed that a statistically significant result is the one for which the "alpha" value is less than 0.05.

While analysing the chi-squared test of independence (chi$^2$), the following hypotheses are assumed:

H0: variables X and Y are independent;

H1: variables X and Y are dependent.

$$\chi^2 = \sum_{i=1}^{r} \sum_{j=1}^{s} \frac{\left(n_{ij} - \widehat{n}_{ij}\right)^2}{\widehat{n}_{ij}} \tag{1}$$

assuming the null hypothesis of the chi-square statistics (chi2) with degrees of freedom

k=(r-1)*(s-1)

where:

r—number of rows;

s—number of columns in the multivariate table.

The following example presents the manner in which the author examined phenomena using the chi-squared (chi2) test.

The chi-squared (chi2) test of independence was applied to show the relationship between the studied characteristics.

Hypotheses for the studied form are of the following form:

H0: variables X and Y are independent;

H1: variables X and Y are dependent.

The statistic is determined from the following formula (chi-square statistic):

$$\chi^2 = \sum_{i=1}^{r} \sum_{j=1}^{s} \frac{\left(n_{ij} - \widehat{n}_{ij}\right)^2}{\widehat{n}_{ij}} \tag{2}$$

Below is presented a multi-divisional table for the examined data:

|  | 1 | 2 | 3 |  |
|---|---|---|---|---|
| <1.39-1.89) | 1 | 3 | 8 | **12** |
| <1.89-2.39) | 0 | 0 | 9 | **9** |
| <2.39-2.89) | 1 | 0 | 6 | 7 |
| <2.89-3.39) | 0 | 0 | 1 | 1 |
| <3.39-3.89) | 1 | 0 | 1 | 2 |
|  | **3** | **3** | **25** | **31** |

The following summaries are helpful for determining the test:

First, we determine the values n$\hat{}$_{ij}:

|  | 1 | 2 | 3 |
|---|---|---|---|
| <1.39-1.89) | 1.161 | 1.161 | 9.677 |
| <1.89-2.39) | 0.871 | 0.871 | 7.258 |
| <2.39-2.89) | 0.677 | 0.677 | 5.645 |
| <2.89-3.39) | 0.097 | 0.097 | 0.806 |
| <3.39-3.89) | 0.194 | 0.194 | 1.613 |

Then, we determine $\dfrac{\left(n_{ij} - \widehat{n}_{ij}\right)^2}{\widehat{n}_{ij}}$:

|  | 1 | 2 | 3 |
|---|---|---|---|
| <1.39-1.89) | 0.022 | 2.911 | 0.291 |
| <1.89-2.39) | 0.871 | 0.871 | 0.418 |
| <2.39-2.89) | 0.154 | 0.677 | 0.022 |
| <2.89-3.39) | 0.097 | 0.097 | 0.046 |
| <3.39-3.89) | 3.360 | 0.194 | 0.233 |

Therefore, the final value of the statistic is:

$chi^2=10.26$.

The critical value is:

$chi^2 alpha= 13.36$.

Thus, it can be concluded that at the significance level alpha = 0.1 $chi^2$ alpha > $chi^2$; consequently, there are no grounds to reject the null hypothesis, i.e., we can assume that there is no relationship between the tested characteristics [80].

## 3. Result

First, the results (Table 1) confirming the overall satisfaction with life of the respondents in the GTA are presented, without taking into account the individual factors influencing the assessment of the quality of life and demographic characteristics of the respondents.

**Table 1.** Overall rating of satisfaction with life in the GTA.

| Level of Satisfaction | I am Very Satisfied | I Am Rather Satisfied | I am Moderately Satisfied | I Am Dissatisfied | I Am Very Dissatisfied | It Is Difficult to Say/I Have No Opinion |
|---|---|---|---|---|---|---|
| Number | 179 | 201 | 91 | 59 | 21 | 32 |

The source: Author's own elaboration based on the conducted research.

Analysing the answers to the question concerning general satisfaction with the place of residence in the GTA agglomeration, the vast majority of the respondents are very satisfied (179 people) and rather satisfied (201 answers). This distribution of responses represents as much as 65% of all the respondents participating in the survey. Only 21 people (3.6%) are very dissatisfied and 59 people (10%) are dissatisfied with the place of residence of the Greater Toronto Area. Obtaining these responses shows that the Polish community in the GTA is one of the ethnic groups that is satisfied with the place of residence.

The author has chosen to analyse several specific elements of satisfaction of the needs in the city, relating to both the economic aspects (accessibility to transport) as well as those related to satisfaction with and access to ecology or aspects related to accessibility to work and housing in the GTA by the inhabitants of the agglomeration. Table 2 shows the results of the survey of the Polonia, assessing the various elements. This analysis applies to all the responses obtained in total, without taking into account the individual demographic and sociological characteristics of the respondents. Assessment of the natural environment was the highest rated factor—181 (31%) people rated it very highly. Accessibility to work (possibility of finding a job) was among the lowest rated elements selected for the analysis. According to 48 (8.2%) respondents, finding a job in the GTA was rated as very bad and 72 (12.3%) rated it as bad, while only 57 (9.7%) rated this aspect as very good, indicating that it was an element which increased the attractiveness of the quality of life in this city.

**Table 2.** General assessment of the selected elements of satisfying the needs of the Polish community living in GTA.

| Element of Satisfying the Needs in the City | Very Good | Good | Average | Bad | Very Bad | I Have No Opinion |
|---|---|---|---|---|---|---|
| Accessibility to housing | 79 | 206 | 129 | 37 | 22 | 110 |
| Accessibility to work—possibility of finding a job | 57 | 74 | 223 | 72 | 48 | 109 |
| Accessibility to transport | 89 | 183 | 187 | 15 | 7 | 102 |
| Assessment of natural environment | 181 | 257 | 80 | 16 | 12 | 37 |
| Accessibility to tourism, leisure, relaxation | 66 | 245 | 161 | 37 | 17 | 57 |

The source: Author's own elaboration based on the conducted research.

The analyses presented below apply to selected items correlated with particular sociodemographic characteristics of the studied respondents. The following demographic features were used for the analysis: age, place of residence of the respondents (Toronto, suburbs) and their length of residence in the GTA. The following statistical analyses also constitute answers to the individual research hypotheses posed in the Introduction of the presented paper.

One of the hypotheses posed by the author assumes that the rating of satisfaction with accessibility to housing increases with age. Table 3 shows the statistical calculations for this hypothesis.

**Table 3.** Age and accessibility to housing.

| Accessibility to Housing | Age | | | | | | | Total |
|---|---|---|---|---|---|---|---|---|
| | 16–19 | 20–24 | 25–29 | 30–39 | 40–49 | 50–60 | Above 60 | |
| Very Good | 8 | 9 | 12 | 18 | 14 | 8 | 10 | 79 |
| Good | 15 | 10 | 26 | 36 | 41 | 40 | 38 | 206 |
| Average | 6 | 37 | 21 | 15 | 19 | 14 | 17 | 129 |
| Bad | 5 | 6 | 9 | 3 | 3 | 4 | 7 | 37 |
| Very Bad | 3 | 2 | 5 | 2 | 2 | 1 | 7 | 22 |
| I Have No Opinion | 42 | 16 | 11 | 6 | 6 | 11 | 18 | 110 |
| Total | 79 | 80 | 84 | 80 | 85 | 78 | 97 | 583 |
| Degrees of Freedom | df = 30 | | | | | | | |
| $\lambda 2$ | 143.82 | | | | | | | |
| $\lambda 2alfa$ | 43.77 | | | | | | | |

Source: Author's own calculations.

The final statistical value is: $\lambda 2 = 143.82$.

The critical value of the chi-square is: $\lambda 2alfa = 43.77$.

At the significance level alpha = 0.05 $\lambda 2$ alpha < $\lambda 2$, i.e., we reject the null hypothesis; therefore, we can say that there is a correlation between age and assessment of accessibility to housing in the GTA. Consequently, the research hypothesis assuming that the rate of satisfaction with accessibility to housing increases with age is true.

The next assumed research hypothesis is that people over 40 years of age assess accessibility to work in the GTA as very good. Table 4 shows the statistical calculations for this hypothesis.

**Table 4.** Age and accessibility to work.

| Accessibility to Work | Age | | | | | | | Total |
|---|---|---|---|---|---|---|---|---|
| | 16–19 | 20–24 | 25–29 | 30–39 | 40–49 | 50–60 | Above 60 | |
| Very good | 8 | 3 | 8 | 7 | 12 | 8 | 11 | 57 |
| Good | 10 | 7 | 9 | 10 | 14 | 11 | 13 | 74 |
| Average | 21 | 36 | 43 | 40 | 26 | 21 | 36 | 223 |
| Bad | 5 | 5 | 7 | 8 | 18 | 21 | 8 | 72 |

| | | | | | | | | |
|---|---|---|---|---|---|---|---|---|
| Very Bad | 1 | 3 | 6 | 8 | 11 | 11 | 8 | 48 |
| I Have No Opinion | 34 | 26 | 11 | 7 | 4 | 6 | 21 | 109 |
| **Total** | 79 | 80 | 84 | 80 | 85 | 78 | 97 | 583 |
| **Degrees of Freedom** | | | | df = 30 | | | | |
| **λ2** | | | | 113,72 | | | | |
| **λ2alfa** | | | | 43,77 | | | | |

Source: Author's own calculations.

The final statistical value is: $\lambda 2 = 113.72$.

The critical value of the chi-square is: $\lambda 2alfa = 43.77$.

At the significance level alpha = 0.05 $\lambda 2$ alpha < $\lambda 2$, i.e., we reject the null hypothesis; therefore, we can say that there is a correlation between age and assessment of accessibility to work in the GTA. Consequently, the research hypothesis assuming that people over 40 years of age assess accessibility to work in the GTA as very good is true.

The research also posed a hypotheses regarding the place of residence of the respondents. The first hypothesis assumes that inhabitants of the GTA suburbs rate accessibility to transport lower than those living in Toronto. Table 5 shows the statistical calculations for this hypothesis.

**Table 5.** Place of residence and accessibility to transport.

| Accessibility to Transport | Place of Residence | | Total |
|---|---|---|---|
| | **Toronto** | **GTA Suburbs** | |
| Very Good | 48 | 41 | 89 |
| Good | 71 | 112 | 183 |
| Average | 57 | 130 | 187 |
| Bad | 6 | 9 | 15 |
| Very Bad | 3 | 4 | 7 |
| I Have No Opinion | 21 | 81 | 102 |
| **Total** | 206 | 377 | 583 |
| **Degrees of Freedom** | df = 5 | | |
| **λ2** | 26, 38 | | |
| **λ2alfa** | 11,07 | | |

Source: Author's own calculations.

The final statistical value is: $\lambda 2 = 26.38$.

The critical value of the chi-square is: $\lambda 2alfa = 11.07$.

At the significance level alpha = 0.05 $\lambda 2$ alpha < $\lambda 2$, i.e., we reject the null hypothesis; therefore, we can say that there is a correlation between the place of residence and assessment of transportation in the GTA. Consequently, the research hypothesis assuming that people living in the GTA suburbs rate accessibility to transport lower than those living in Toronto itself is true.

Another hypothesis relating to the place of residence of the studied population (Toronto or suburbs) assumes that people living in the GTA suburbs rate the natural environment more highly than those living in Toronto itself. Table 6 shows the statistical calculations for this hypothesis.

**Table 6.** Place of residence and assessment of the natural environment.

| Assessment of the Natural Environment | Place of Residence | | Total |
|---|---|---|---|
| | **Toronto** | **GTA Suburbs** | |
| Very Good | 67 | 114 | 181 |
| Good | 112 | 145 | 257 |
| Average | 5 | 75 | 80 |

| | | | |
|---|---|---|---|
| Bad | 5 | 11 | 16 |
| Very Bad | 3 | 9 | 12 |
| I Have No Opinion | 14 | 23 | 37 |
| **Total** | 206 | 377 | 583 |
| **Degrees of Freedom** | df = 5 | | |
| **λ2** | 38,27 | | |
| **λ2alfa** | 11,07 | | |

Source: Author's own calculations.

The final statistical value is: $\lambda 2 = 38.27$

The critical value of the chi-square is: $\lambda 2alfa = 11.07$

At the significance level alpha = 0.05 $\lambda 2$ alpha < $\lambda 2$, i.e., we reject the null hypothesis; therefore, we can say that there is a correlation between the place of residence and assessment of the natural environment. Consequently, the research hypothesis assuming that people living in the GTA suburbs rate the natural environment better than those living in Toronto itself is true.

The research also examined the duration of residence of the respondents in the studied area. One of the hypotheses assumes that people living in the GTA for more than 20 years rate accessibility to tourism, leisure and relaxation lower than those living in the GTA for a period shorter than 20 years. Table 7 shows the statistical calculations for this hypothesis.

**Table 7.** Duration of residence and accessibility to tourism, leisure and relaxation.

| Accessibility to Tourism, Leisure, Relaxation | Duration of Residence | | | | | | | Total |
|---|---|---|---|---|---|---|---|---|
| | Since Birth | Several Months- a year | 1–3 years | 4–10 | 11–20 | 21–30 | Above 30 | |
| **Very Good** | 6 | 2 | 1 | 1 | 7 | 26 | 23 | 66 |
| Good | 78 | 2 | 7 | 9 | 56 | 37 | 56 | 245 |
| Average | 23 | 2 | 5 | 2 | 14 | 89 | 26 | 161 |
| Bad | 11 | 1 | 3 | 0 | 1 | 13 | 8 | 37 |
| Very Bad | 12 | 0 | 1 | 3 | 0 | 0 | 1 | 17 |
| I Have No Opinion | 12 | 5 | 6 | 6 | 9 | 10 | 9 | 57 |
| **Total** | 142 | 12 | 23 | 21 | 87 | 175 | 123 | 583 |
| **Degrees of Freedom** | df = 30 | | | | | | | |
| **λ2** | 176,49 | | | | | | | |
| **λ2alfa** | 43,77 | | | | | | | |

Source: Author's own calculations.

The final statistical value is: $\lambda 2 = 176.49$

The critical value of the chi-square is: $\lambda 2alfa = 43.77$

At the significance level alpha = 0.05 $\lambda 2$ alpha < $\lambda 2$, i.e., we reject the null hypothesis; therefore, we can say that there is a correlation between the duration of residence in the GTA and assessment of accessibility to tourism, leisure and relaxation. Consequently, the research hypothesis assuming that people living in the GTA for more than 20 years rate accessibility to tourism, leisure and relaxation lower than those living in the GTA for a period shorter than 20 years is true.

## 4. Discussion

Inhabitants of the cities decide on the set of needs that are concentrated in the urban space and result mainly from the basic functions of this kind of settlement unit. In order to survive, cities must offer their inhabitants the possibility of existence in their area, i.e., provide the following: water, food, clothing, housing, jobs, living services, transport, security. Additionally, they must create the conditions for social contact, most often

through the designation of public facilities and spaces [81,82]. Urban exchange space can have a different intensity of usage by users (residents and visitors), which is determined by its attributes (equipment, organisation, values) and the characteristics of the city (location, genesis, size, economic position, social structure).

One of the most important observable and universal demographic factors influencing the assessment of perceived quality of life is age [83]. As Patterson and Pegg [84] point out, retirement today is increasingly a time when seniors begin to experience the feelings of freedom, including taking risks or attempting things that they could not have done before, due to their work and family responsibilities, and thus begin to enjoy the urban space they have chosen to live in.

Sociological analyses attach great importance to age since it determines personal social roles, health, responsibilities as well as assessment of urban space according to the age stage of this person. Satisfaction of the basic living standards in the human production phase, such as having a house and a job, is very important from the point of view of assessing urban space. Cities that offer the chances of finding a good job attract new inhabitants who need a place of residence at the moment of undertaking employment. The aspect of housing availability was taken into account while conducting the research. Income levels and housing prices constitute the basic measures to identify areas inhabited by various demographic and socioeconomic groups and—as a result—with different needs and problems. Nical Senlier and Reyhan Yıldız [85] carried out research in Kocaeli, Turkey that confirmed that seventy-six percent of the survey participants indicated that it was impossible to find good housing at a reasonable price and that competitive property prices in the cities attract new residents and workers to agglomerations. While comparing Kocaeli with European cities, the Turkish city still has more opportunities to find housing than Copenhagen, Munich, Helsinki, Vienna or Rotterdam. The hypothesis put forward in the presented paper has been proven, i.e., as a person gets older, the rating of satisfaction with accessibility to housing increases. This is connected with the fact that the older a person gets, his or her work experience, internships and work competencies increase, which leads to better earnings, thanks to which accessibility to housing automatically becomes easier. These observations have been confirmed by another hypothesis that states that people over the age of 40 rate accessibility to work in the GTA very highly. However, the research conducted by Nical Senlier and Reyhan Yıldız proves that the process of employment and looking for a job is not easy in cities and that unemployment rates are higher in cities as compared to suburban areas. It is observed that in the cities of Helsinki, Stockholm and Brussels, employment opportunities are relatively better in comparison to other cities, e.g., Lisbon [85]. The research showed that European cities of a similar size to Kocaeli, Turin, Helsinki, Rotterdam, Copenhagen, Stockholm, Brussels, Munich, Vienna and Marseille are comparable when taking into account the quality of urban life with regard to job opportunities.

Quality of life in the city is influenced by a variety of factors of diverse importance—these include health, safety, access to services (including public services), opportunities for rest and recreation, quality and friendliness of a space (including, in particular, public spaces—streets, squares, squares), cost of living, transport efficiency, occurrence of pollution (particularly air pollution and noise) and functioning in the community [28,32,33,35–38,86–88]. The manner of shaping transport in the city has a larger or smaller impact on all the above-mentioned elements. Sometimes, this influence is direct; however, most often, it is indirect. This results from the fact that transport allows people to satisfy a variety of needs outside the house [89]. Thus, in this situation, it is the satisfaction of life needs that will directly influence quality of life. However, it is necessary to reach the places where these needs are met in order to make this happen. One of the research hypotheses has confirmed the posed assumptions that people living in the GTA suburbs rated accessibility lower than those living in downtown Toronto.

Inhabitants of the city centre who make use of the fixed assets usually located in the city centre and spend their daily lives in this fragment of the urban space perceive its

characteristics and—at the same time—influence its intensity or aesthetics with their consumption behaviour. The details conditioning the use of the numerous public spaces situated in this part of the city are important to them and attract various city users to the city centre. The inhabitants of the city centre become representatives of the characteristic features of the city, mainly towards visitors, identifying the style of development with their preferences. Simultaneously, their needs are met primarily in the city–environment system. What is most valuable in the city for the inhabitants of the centre is usually in the immediate vicinity, while obtaining what they lack requires their departure from the city, at least going to the suburbs (shopping centres, leisure) as well as more distant places. People living in the city appreciate the proximity of their workplaces, saving on transport costs and commuting time. However, the ability to move from place to place constitutes an important aspect of city life. Efficient transport saves time and has a positive impact on the satisfaction of residents, while inefficient transport can be a source of frustration and stress [90]. Active transport is also less disruptive to the environment and other residents, which is why many cities are trying to promote it through the creation of infrastructure and various promotional activities. Therefore, spatial variation in the quality and accessibility of public transport and conditions for active travel constitutes a significant aspect of the study of the quality of life in the cities, which has also been examined and included in the research conducted on the Polish community in the GTA. Residents of Toronto rated public transport highly, while, e.g., only 2% are satisfied with public transport in the city of Kocaeli. Among the respondents surveyed in the GTA, 65% are suburban residents who commute to central areas of the city and focus their settlement preferences on the benefits of the natural environment and suburban lifestyle. Their subsistence needs are met equally in the immediate vicinity of the house and in the inner-city trade and service facilities. However, they satisfy higher-order needs close to the house to only a small extent, which forces them to commute frequently to schools, cultural institutions, healthcare facilities and dining and entertainment venues. The constant movement and the necessity of using public transport, which is less developed in the suburbs than in the inner city, is probably the reason that the Polish community living in the suburbs of Toronto has a poorer assessment of the public transport in the GTA.

More than 27 million Canadians, or seven out of ten Canadians, live in large urban areas. However, more and more people are moving out of Canada's large cities to the suburbs and into smaller towns that are continuously growing. The trend began even before the 2019 pandemic and COVID-19 accelerated such decisions in many cases. An urban and real estate economist at Ryerson University in Toronto, Frank Clayton, believes that the pandemic has only increased the exodus that has been taking place for several years. He also pointed out that millennials, the generation of people born in the last two decades of the 20th century, who are just now starting families and having young children, are repeating decisions previously made by those born between 1949 and 1964, known as the baby boomers. They also fled the large cities, moving to the outskirts.

Statistics Canada data show that, for example, in a number of approx. 160,000 residents in Oshawa, which is part of the GTA, where the author's study was also conducted, population growth was 2.1 percent during the study period, in part because of moves from the Greater Toronto Area. The Kitchener–Cambridge–Waterloo area, northwest of Toronto, saw a similar increase. Numbering more than 110,000 residents as recently as 2016, the Milton area in southern Ontario is one of Canada's fastest-growing urban centres. During the period studied by Statistics Canada, population growth there was 4 percent [91]

An urban and real estate economist at Ryerson University in Toronto, Frank Clayton, believes that the pandemic has only increased the exodus that has been taking place for several years [92]. He also pointed out that millennials, the generation of people born in the last two decades of the 20th century, who are just now starting families and having

young children, are repeating decisions previously made by those born between 1949 and 1964, known as the baby boomers. They also fled the large cities, moving to the outskirts.

Other scholars who have highlighted the decrease in population in the city centre and its increase in the suburbs are Barreira et al. [93], who, in their article, highlighted that population growth in European and North American cities accelerated in the 19th century and was most stabilised in the mid-20th century, and others continue to urbanise, as in the case of Portugal. However, due to declining fertility rates, most European countries are likely to experience population decline in the near future, increasing the likelihood of a broader dichotomy between suburbs gaining population and cities losing population. They also point out that economically successful cities attract new residents, which is often associated with an increase in the quality of urban life, and with the additional provision of affordable housing and a friendly environment, such measures can help to attract new residents. Another article by Barreira et. al. [94] notices that the number of cities affected by population decline is increasing worldwide. The topic of population decline has been seen by politicians as a sign of ineffective governance; on the other hand, places that lose residents can be seen as an opportunity to gain a better lifestyle for those who stay in them, as such losses can reduce urban crowding, especially during transportation rush hours.

Urban population decline is often thought of as a condition that leads to disorder, decay and, as a consequence, unhappiness or poor quality of life. However, empirical evidence has shown that this is not always the case [95]. Cities, while losing population, gain quality of life because they are less crowded, less polluted and have more open spaces [96,97], thus contributing to housing satisfaction and a lower need for citizens to move out. According to Amérigo [98], three main evaluative aspects should be considered when assessing housing satisfaction: special (architectural and urban features), human (socio-relational features) and functional (services and amenities). Spatial aspects refer to residents' perceptions of the environmental quality of where they live, with features such as pedestrian walks, roads and quality of outdoor facilities being of particular importance [99], as well as urban design/layout, crime rates and housing quality [100]. Human factors include social connection to a place and civic engagement. Building social ties plays a key role in housing satisfaction because it promotes in residents a sense of trust, mutual help and psychological comfort that translates into high levels of place attachment and housing satisfaction. Functional factors refer to the availability of education, police services, public transportation, parks and green spaces, as well as leisure opportunities, space for sports and time spent on daily commuting to and from work [101].

The place of residence and assessment of the environment have a very significant impact on the overall rating of the quality of life in the city as well. The research confirms that people living in the GTA suburbs rated the environment more highly than those living in Toronto itself. Environmental definitions are related to access to a specific quality of the natural environment and the broadly understood socioeconomic environment, such as "quality of life includes individual satisfaction, material well-being, an ecologically healthy environment and the opportunity to shape the individual lifestyle or integrate the individual into society" [102] or "quality of life is the individual's perception of his or her position in the natural environment in which he or she lives, assessed through the prism of the goals adopted by him or her, the current condition of physical and mental health, his or her independence and beliefs as well as the prevailing social relations" [103].

Many people choose the suburbs for peace, quiet and the availability of greenery in the form of their own garden or open spaces and woodlands [104]. The desire to maintain access to nature is justified by the benefits proven by studies [105,106], confirming that staying among greenery helps to reduce stress, relax and restore the ability to concentrate, which translates into a high rating of the quality of urban space. The ecological domain constitutes an important element of a good place in the quality of life. Murgaš

and Klobučník [107] found a correlation between the balance of emissions and the quality of life index, proving that people living in the suburbs of large cities evaluate the environment and ecology in the context of their quality of life more highly than those living in city centres. However, the city of Toronto has been described as a "city in the park". With over 1500 parks (over 8000 ha), Toronto's green space inventory is extensive [108]. Over 12% of the urban area of Toronto consists of green spaces administered by the City of Toronto and with the Toronto Region Conservation Authority; 71% of the areas were classified as natural heritage areas (valley areas, ravines, densities of trees, paths along the waterfront, etc.).

The research by Zhonghua Gou, Xiaohuan Xie, Yi Lu and Maryam Khoshb [109] was conducted in Hong Kong, where living conditions have become a major factor influencing people's quality of life. The environment was the most influential factor with regard to the overall quality of life in this city. The research also confirmed that different groups of people have various needs for residential environments: the low-income group needs better location and privacy whereas the middle and high-income groups require better architectural quality rather than a clean environment.

Duration of residence in a specific area also seems to be important in assessing the quality of urban space. The longer the person lives in the same area, the better the person knows it and enjoys its attractions, opportunities and offerings. In assessing the level of personal satisfaction with the general standard of living, it seems necessary to take into account leisure and relaxation, which are indispensable for happy functioning. Therefore, in the conducted research among the Polish community, the length of participants' residence in the GTA and assessment of accessibility to tourism, leisure and relaxation were correlated. The results proved that people living in the GTA for more than 20 years (but not since birth, so the results suggest that older people answered this way) rated accessibility to tourism, leisure and relaxation lower than people living in the GTA for a period shorter than 20 years. This may result from the fact that people living in the GTA for a long time have often visited places intended for relaxation and leisure, which have become unattractive to them over time. However, global data confirm that tourism activity increases with age. Economic development and medical advancement have made the present pensioners the healthiest and wealthiest seniors in the history of mankind and they are characterised by higher levels of tourism activity than ever before [110]. This is confirmed by the research conducted in the European Union states (*Facts and Figures: The European on Holidays 1997-1998*), data on the tourism activity of the older residents of the USA as well as the research conducted in other research centres [111,112].

In studies on the tourism activity of senior citizens, the vast majority of researchers use the traditional criterion of metric age. The studies of this kind have shown that the level of tourism activity of people over 55 initially increases until they reach the age of around 65-68, and then it decreases [113,114]. Furthermore, younger seniors choose more active activities during tourist trips; however, preferences change as they age. Older tourists—as their life experience increases—are more interested in leisure, environmental values, interpersonal contacts as well as local cuisine and handicrafts. They also pay more attention to safety and a high standard of hygiene. In Spain, it has been observed that the average length of a tourist stay for people over 50 is 6% higher, and over 60, it is even 25% higher than for people under 50 [115].

In the case of Toronto, we deal with the situation when residents and visitors to the city share the same institutions and facilities, including recreational areas. The integrated coexistence of both groups of users is undoubtedly facilitated by the existing organisation of the space of this city. Its leading feature is the "egalitarian" urban layout: quarters of similar size and layout, delimited by a regular rectangular grid of streets and considerable dispersion of numerous objects, constituting a resource of contemporary cultural tourism in this layout [54]. This is definitely different from the situation in the European cities, where an individual can encounter a conflict situation between two groups of users of urban exchange space: residents and newcomers, where the latter are clearly

treated as a collective endowed with negative attributes. The negative attitude towards the successively increasing number of tourists is reflected in the concept of overtourism, which is increasingly used to describe this situation, not only by researchers but also by city managers [116–120]. The research from Liberec, Slovakia [37] reports very little difference between women and men in satisfaction with quality of life in the city, and quality of life and satisfaction with accessibility to recreational places in the city increase with age. The oldest residents and those living with Liberec the longest are the most satisfied. In this context, the situation in Toronto is different, which, however, is not reflected in the results obtained among the Polish community living in the GTA for more than 20 years.

Toronto was chosen as a study area because it provides a unique opportunity for research on ethnic minorities—it is the most important immigrant-receiving city in Canada. Toronto attracts around half of all immigrants who come to Canada. Therefore, when analysing the quality of urban space by Polish people in Toronto, in addition to the analysis and examples of assessing urban quality in the world, it is worthwhile to present selected studies conducted on different ethnic groups assessing the quality of urban space in the described city.

Carlos Teixeira [121], in his research, performed an analysis between two ethnic groups living in Toronto, between the Portuguese and the black race. In his research, he presented the strategies and barriers that businesses owned by representatives of ethnic groups face, in order to assess how ancestry affects running a business in the city. The research indicated that Portuguese people differed significantly from black entrepreneurs in that they were more likely to rely on their community ("ethnic") resources. Black entrepreneurs faced more barriers to starting and/or running their current business, particularly in obtaining loans from financial institutions and banks in Toronto, which contributed to their overall poorer assessment of the quality of urban spaces.

Jennifer Logan and Robert Murdie [122] performed a description of Tibetan refugees settling in Toronto, with a particular focus on whether they were able to obtain affordable homes quickly. The study showed that most Tibetans have had a home in Toronto for less than ten years, and their overall assessment of the quality of urban space is poor.

Meanwhile, Mohammad Quadeer, Sandeep K. Agrawal and Alexander Lovell [123], in their research, focused on describing four ethnic groups living in Toronto, and sought to assess their satisfaction with living in the city. They chose Chinese immigrants, Jews, Portuguese and Italians as their research group. Their research found that migrant enclaves are mainly located in suburban areas, where the percentage of homeowners is high, and their satisfaction with the location and quality of life is high.

In his article, Kenneth L. Dion [124] described the Housing New Canadians project, in which he presented the results of a study on immigrants' perceptions of discrimination in the search for rental housing. In his study, respondents from three immigrant communities—Jamaican, Poles and Somalis—indicated the perceived housing discrimination that they experienced personally and how the discrimination was directed at their migrant group, with Poles rating this aspect much less negatively than the other study groups. Respondents also rated the extent to which their nationality, income level, source of income, immigration status, English language proficiency, ethnicity, religion and family size contributed to their negative assessment of the discrimination and prejudice that they perceived. Jamaican and Somali immigrants perceived more personal and group discrimination than Polish immigrants, who, in the survey, reported overall satisfaction with their standard of living in Toronto.

Kenneth L. Dion and Kerry Kawakami [125] of the University of Toronto analysed the quality of life ratings of six ethnic groups, representing minorities (Black, Chinese and Asian) as well as white minorities (Italians, Jews and Portuguese). The study found that Black, Chinese and Asian participants felt more discrimination against their group than white minorities, especially in economic areas of life regarding finding a job, salary

and social advancement, thus rating urban space lower in these aspects than other study groups.

Drawing on translational theory and previous research on the housing trajectories of new Canadians, Sutama Ghosh [126] analysed the housing experiences of two "South Asian" groups in Toronto—Indians from India and people from Bangladesh. Highlighting important differences among the intra-immigrant groups, the study showed how diverse transnational ties influence their choice of neighbourhood and housing type and quality. Indians are dispersed throughout Toronto, while Bangladeshis have gathered in specific parts of the city, often having smaller houses and being less satisfied with life in Toronto than Indians.

Toronto is a major immigrant-receiving city in Canada and contains a wide diversity of ethnic groups. Although Canadians are generally open to immigration, there is evidence that some recent immigrant groups, especially those concentrated in the suburbs of Toronto, do not do well with social integration, which translated into their assessment of the quality of life in the city. This is demonstrated by Murdie R and Ghosh S [127] in their research. Their findings challenge traditional perspectives on ethnic concentration, particularly the spatial assimilation model, and highlight the importance of considering subjective integration, especially life satisfaction in the new country, as a means of mitigating barriers to poor functional integration. In the case of Toronto, spatial concentration does not necessarily imply a lack of integration, although, in the case of Asian immigrants, who tend to concentrate in suburban enclaves, their withdrawal from urban life is clearly evident.

Kant et al. [128], who published data on the quality of life of aboriginal people in Canada in 2013, are well known. The research team conducted these surveys in the provinces of Ontario (population 600, 120 households) and British Columbia (population 1500, 275 households). During the study, 316 questionnaires were collected. The questionnaire included questions about satisfaction with general well-being, education, employment, health, housing, income and land use. The research showed that the Aboriginal residents are not satisfied with their health or quality of life. The research indicates that the development of a national policy based on attention to mental health in Aboriginal residents, and assistance from the government in this area, would significantly increase the quality of life for these populations in Canada.

Another study on the quality of urban living space was conducted by Wei-Wei and Alicia Garcia [129] on the Chinese diaspora. The study investigated the sociocultural adaptation and change in settlement quality of life among recent elderly Chinese immigrants in Canada. The study was conducted on 31 elderly men and women who had recently immigrated from China to Canada, and their main purpose was to explore their perceptions of quality of life in a new place. Their overall perceptions of post-immigration quality of life were marginalised by losses and gains, and depended on multifaceted factors such as language, intergenerational relationship, economic status and their support systems.

The assumption of the research conducted on the Polish community in the GTA was to construct a model based on which it was possible to examine the Polish community in many areas of life, concerning the level and quality of life (in such aspects as cultural, social, economic, educational, legal, geographical and health). This is a novelty that the author proposes as a research tool to assess the quality and standard of living of the population. The research results presented in this article are only concerned with selected indicators that are used for the general assessment of urban space. The article does not describe the model constructed by the author, nor does it provide a detailed analysis of individual factors, but the constructed model and the results of the research conducted on the Polish community of the GTA provide the possibility for further, reliable and in-depth analysis of quality of life in such areas of life as economic, cultural, social, educational, legal, geographical and health.

## 5. Conclusions

Agglomerations are changing and populations increasing at a rapid pace. Facilities for people with disabilities used to constitute determinants of whether a public space was friendly or not. Today, this is not enough and the very definition of a place that can be considered sustainable, open and functional is still not clarified. However, it is possible to identify good design practices and directions for the development of friendly spaces. Currently, the most successful cities are those which have addressed the needs of their inhabitants and have understood that the means to achieve friendly spaces is to design urban infrastructure in such a way that the inhabitants and visitors to the city can coexist. Consequently, the model concepts include more pedestrian facilities and solutions, such as parklets, woonerfs, community gardens and urban farms. Thus, the role of small architecture in public spaces is increasing and there are more and more green regions as well as open functional areas.

Taking into account the multiculturalism of Toronto, the examination of such a large ethnic group as the Polish community (that has retained its cultural individuality and identity, apart from assimilation with other minority groups in the GTA) should constitute the reason for continuation of the research on the quality of life among other ethnic groups. The results of the conducted surveys concerning quality of life should be made available to city office employees, city councillors and housing estate councillors and provide a starting point for spatial planning, transport planning, management of urban greenery and public spaces as well as revitalisation measures. The diagnosis should also be available to non-governmental organisations and socially active citizens so that it can become the foundation for public discussion and public participation in decision-making on budget expenditure during public consultations and submission of projects to civic budgets. As a result, regular surveys regarding quality of life can constitute an important part of the knowledge-based and participatory decision-making process in the cities.

To improve the quality of life for residents in the GTA, the City of Toronto, along with its suburban towns, should join together in a global network, using the potential of Internet communication. They should unite their forces and work together to counter growing problems, help each other to respond to crises, disasters and threats, and be inspired by and learn from each other. Cities should combine elements of local tradition with global trends and should also be open to outside influences. Actions by local governments in Toronto that could contribute to a better assessment of the quality of urban space by both its residents and visitors to the metropolitan area could be the following:

- Lowering land taxes and housing prices. The lack of housing or the existence of only houses that are too expensive promotes urban sprawl into suburbs, thus reducing the urban population.

- Increasing the abundance of urban transportation. A larger public transit fleet will result in even more people working in Toronto using it, which will contribute to residents not having to use their own mode of transportation (car). Such an action will result in less vehicle traffic in the downtown area, less traffic congestion and less exhaust pollution. Investment in developing mass transit so that it is easily accessible, well connected to building entrances and pedestrian routes.

- Implementing high-tech industries that are environmentally friendly and that lead to higher efficiency, as opposed to heavy industry. These technology-based industries, which are associated with promoting improved quality of life, should attract more workers and residents to cities.

- Local governments should pay attention to the street space, which should feature an appropriate amount of diverse functions, including retail and service, which includes establishments that are open late in the evening (stores, bars and restaurants).

- In urban planning, in order to improve the quality of life in the city, it is worth paying attention to the location of housing developments and workplaces next to each other, with good accessibility, within a ten-minute walking distance, to basic public and

commercial institutions, recreation and sports areas and playgrounds, squares and attractive public spaces.

- In city parks and squares, there should be places to sit and lie down, viewpoints, places for larger groups to meet, picnic tables and playgrounds and even places for computer work.

- It is worth paying attention to publicly accessible places that enable active recreation in public space, as opposed to fitness clubs and gyms. Urban recreation is for everyone, regardless of age, social status, wealth or disability, for people who practice sports professionally and for amateurs. Easy access to parks, green areas and river banks gives the opportunity for residents to spend time actively.

- The enrichment of urban infrastructure with jogging tracks, bicycle paths and skateboard routes, in addition to encouraging physical exercise in leisure time, can also be used as an alternative way to travel to work and other places.

- The organisation of festivals, feasts and events by local authorities in public spaces. Such activities strengthen the connections between people and between places and people. By living, working, resting and playing, people can build connections and cultural heritage.

**Author Contributions:** All figures, tables, calculations, methodology, software, validation, formal analysis, investigation, resources and writing were made entirely by the author. The author have read and agreed to the published version of the manuscript.

**Funding:** This research received no external funding.

**Data Availability Statement:** Data available in a publicly accessible repository.

**Acknowledgments:** Author offers thanks to the reviewers for all comments and suggestions. They contributed significantly to the current article.

**Conflicts of Interest:** The author declares no conflict of interest.

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
