# Peer review of "Satisfaction with Selected Indicators of the Quality of Urban Space by Polonia in the Greater Toronto Area"

_land, doi:10.3390/land10080778_

Round 1
Reviewer 1 Report
The paper has been improved.
It is matching the Land topics as part of a special issue.
The applied methodology is generally clearly explained. Nevertheless, there is no need to be so detailed as the applied methods are well known and simple.
Author Response
Response to Reviewer 1
Thank you very much for the received review. I am very grateful for all the comments, guidance and your time.
Once again, I would like to express my great gratitude for all the recommendations that have been pointed out by the reviewer, which made the quality of the article and its structure clearer and richer in content.
Response 1:
Obviously, the author agrees with the reviewer that the methods used in the article are familiar, but given the suggestions of another reviewer, the point was expanded upon. The methodology used in the article was described in such detail because one of the comments of another reviewer was that the methodology was described insufficiently and unclearly. Therefore, the author expanded this issue responding to the comments of another reviewer who had accepted this expanded version. Nevertheless, the author agrees with the reviewer that the methods used in the article are known and generally applied.
Thank you very much for the advices and suggestions.

Reviewer 2 Report
While I still believe that this article would be a better fit for Sustainability, changing the title and emphasizing the space phenomenon in the article has improved publishing opportunities in Land. Which, while maintaining a positive assessment of the research and support in the literature on the subject, prompts me to accept the content in this form. However, I would consider shortening the last part to conclusions that result directly from the research carried out.
Author Response
Response to Reviewer 2
Thank you very much for the received review. I am very grateful for all the comments, guidance and your time.
Once again, I would like to express my great gratitude for all the recommendations that have been pointed out by the reviewer, which made the quality of the article and its structure clearer and richer in content.
Thank you very much for accepting my article and recommending it.
Response 1:
According to the suggestion of the reviewer, the last part of the article, where the author gives the conclusions directly resulting from the conducted research, has been removed.
Thank you very much for the advices and suggestions.

This manuscript is a resubmission of an earlier submission. The following is a list of the peer review reports and author responses from that submission.
Round 1
Reviewer 1 Report
Dear author,
Thank you for sharing with me your paper titled:
Evaluation of the Quality of Urban Space by Polonia in the Greater Toronto Area (GTA)
The paper is rich in information and it seems that the author is familiar with the relevant literature for her work. However, I found that the promise in the title was not addressed. Urban spaces are hardly analyzed in the paper but other topics.
I was not convinced that there is any connection between the Polish origin of the informants and the findings. Why were they chosen and not others? 4-6% of the GTA is not a dominant group. Is it the biggest population of the same origin ??
Introduction
"The city as an anthropogenic product constitutes a natural environment"
This phrase is very problematic. City is not a natural environment and especially from an anthropogenic perspective. In the following there is a lot of information, but not a clear discussion of what is urban space and in relation to the contemporary multicultural city.
It seems that the paper deals with quality of life and I wonder why did the author chose these variables and not others to explore.
Method and result
The method is fine. However, I am not sure that such a detailed description is needed. Currently there are more advanced representation methods to present results in a more vivid and communicative way.
The findings could be interesting if positioned in a relevant framework. (comparing to other minorities or comparing new comers to old ones etc. and situating the results within the cultural sphere.
Discussion
The comparisons are not clear. Why did the author chose Kocaeli for comparison? why Hong Kong (line 509)? (Living conditions determine quality of life almost everywhere) etc.
Please refer to the number of new comers to Canada each year 390,000 or a quarter of Million.
In conclusion,
I recommend re-writing the paper around the findings and their meaning.
Sincerely yours,
Author Response
Response to Reviewer 1 Comments
Point 1: I was not convinced that there is any connection between the Polish origin of the informants and the findings. Why were they chosen and not others? 4-6% of the GTA is not a dominant group. Is it the biggest population of the same origin ??
Response 1: Contemporary distribution of Poles in Canada was conditioned, on the one hand, by the processes of development of the young state and, on the other hand, by social changes in the Polish community itself. Until the 1930s, immigrants from the Central Europe were directed to agricultural areas in the provinces of Manitoba, Saskatchewan and Alberta by Canadian authorities. It was only after the Second World War that Poles began to settle mainly in cities. Urbanisation processes resulted in the fact that as early as in the 1970s, 80% of Poles lived in cities, mainly in the large agglomerations of the provinces of Ontario and Quebec. Nowadays, cities, particularly Toronto and subsequently: Montreal, Ottawa, Edmonton, Hamilton, Vancouver, Winnipeg, Calgary constitute the main centres of the Polish community. Representatives of the most recent waves of the Polish migrants from the second half of the 20th century settled mainly in Toronto, hence the strong numerical dominance of the Polish community living there. The most active organizations operate in this city and most of the Polish press titles are published here (Reczyńska, 2006). 35,151,728 people lived in Canada in 2016, where 1,106,585 inhabitants were of Polish origin. This is 3.9% of the total population. Ontario, where Greater Toronto Area is located, constitutes the most numerous province inhabited by Polonia. 523,490 people lived in Ontario in the year 2016, while the largest number of people of Polish descent in relation to all residents is in Manitoba, where the percentage of Poles is as high as 6.9%.
259,715 people of Polish origin live in the Greater Toronto Area, which is 4% of the total population in this area. Most Polish people live in Mississauga, in the county of Peel. This population equals 43,350 people, which is 6% of the total population in Mississauga. A significant number of people of Polish origin live in the county of York, in Vaughan. This number includes 18,265 people, which is 5.96% of the total population in the city. The smallest number of Polish people lives in two towns in the county of Durham. This is a city of Brock, with 335 people of Polish descent, and a city of Scugog, with 590 people.
In 2016, Statistics Canada published census data which confirms that 34 million Canadians use over 200 languages on a daily basis. In the table of Statistics Canada (2016), Polish is the eleventh most widely spoken language in Canada.
According to Reczynska, Soroka (2013), in 1991, Poland was on the second place (after Hong Kong) in the summary of ten countries constituting the most important sources of immigration to Canada. In 1996, Poland shifted to the sixth place (after Hong Kong, China, India, the Philippines and Sri Lanka), however, it was still ahead of other European countries. The last population census, including persons declaring Polish origin, is available from 2016. At that time, 259,715 people declared Polish origin in the area covered by the survey (GTA), which is 4% of the share of the Polish community to the number of all GTA residents ( 6,417,516). Such a large number of the Polish diaspora in the GTA, clearly demonstrating its distinctiveness in the multicultural Canada, constitutes an important topic for exploring and information for sociological research concerning the population in Canada. Taking into account the fact such a large group lives not only in the city of Toronto, yet, within the entire GTA area, the research was also conducted outside Toronto itself.
There are familiar publications of Canadianists who raise an aspect of history, multiculturalism, migration, Polish pastoral activity as well as causes of emigration and Canada's policy towards people who obtained the status of emigrants in their studies. The topic and research taking into account the quality and standard of life of the Polish community in Canada in the context of quality of urban space undertaken by an author would fill the gap in the research works on the Polish diaspora in Canada. The research complements the historical threads concerning emigration from Poland in the context of the quality of life of the current Polish community living in the Greater Toronto Area. A large wave of Poles emigrating to Canada took place in the 1960s and 1970s. Today, the people who arrived at that time are the ones who are 60 years old or older. They are often people after their professional activity, currently retired, often requiring medical care and evaluation of the quality of urban space has a significant impact on their assessment of the general quality and standard of living.
In the conducted study, one of the questions concerned the sense of Polish national identity. Respondents were asked if they feel Polish. 97.4% of the respondents answered ‘yes’. Only 0.6% of the respondents answered that they did not, and 2% did not have an opinion on the subject.
Point 2: "The city as an anthropogenic product constitutes a natural environment"
This phrase is very problematic. City is not a natural environment and especially from an anthropogenic perspective. In the following there is a lot of information, but not a clear discussion of what is urban space and in relation to the contemporary multicultural city.
Response 2: The sentence was chenged from „The city as an anthropogenic product constitutes a natural environment” to „The city as an urban space area where…..”
Point 3: It seems that the paper deals with quality of life and I wonder why did the author chose these variables and not others to explore.
Response 3: The quality of life in the city is affected by a variety of factors of varying importance – these include: the possibility of rest and recreation, the quality and friendliness of space (including especially public spaces – streets, squares), the cost of living, access to work and housing, the efficiency of transport in the city, as well as the presence of pollution, especially air pollution and noise (Węziak-Białowolska, 2016, Kopeć, 2019). Taking, among others, these factors, listed by scientists, into account, the author, while presenting the research in the following article, focused on the respondents’ evaluation of similar factors listed in scientific works affecting the evaluation of urban space, such as: accessibility to housing, possibilities of finding a job, public transport, accessibility to recreational tourism in the city, and quality of the natural environment. All these factors significantly affect the quality and evaluation of urban space. The author wanted to focus and examine how the very place of residence within the GTA influences its evaluation by the respondents, so she first wanted to see what the residents’ satisfaction is with the very accessibility to real estate (housing). An important factor in any city is the availability of work, which allows one to receive proper salary, which in turn provides a good and decent life, as well as allows one to fulfill one’s hobbies and enjoy the wide range of cultural, educational, entertainment, tourist and recreation offer provided by the city. Geographical factors largely contribute to the quality of life in any city. Therefore, the author included in the study both the factors of moving around the city (public transport), as well as the place of residence in the GTA itself, and access to tourism, understood as tourist area in the geography of tourism.
Point 4: The method is fine. However, I am not sure that such a detailed description is needed. Currently there are more advanced representation methods to present results in a more vivid and communicative way.
Response 4: The author believes that thorough and reliable description of the methodology used in the presented research is necessary, so that the reader can accurately understand the scheme of the study, and also in the future can use a similar scheme in their research on another research group.
Point 5: The findings could be interesting if positioned in a relevant framework. (comparing to other minorities or comparing new comers to old ones etc. and situating the results within the cultural sphere.
Response 5: In order to make the research as representative as possible, seven age groups have been differentiated, with the use of the pre-conceived division. These were the following age groups: 15-19, 20-24, 25-29, 30-39, 40-49, 50-59, over 60 (where women and men were additionally included separately). Conducting research among people aged 15 and over (youth) allowed to be sure that respondents understood the asked questions and answers provided by them were reliable. In this article the research results are presented for all age groups together.
The questions applied in the questionnaire were formulated in such a way as to refer to the experiences of the respondents in 2018 and 2019 as well as from the period of the last several years. This approach, according to the author, has enabled to capture average assessment of the quality and standard of living of the surveyed group of the respondents. During the analysis of the quality and standard of living, factors, such as: gender, age, education, profession, achieved material level, size of the household, regulated residence status were used. This has allowed for a more precise description of the examined group and identification of the relations between interesting aspects.
However, the purpose of this article was not to indicate the quality of living standards in particular age groups, but to capture the statistical relationship in the surveyed population of Polish diaspora, and to indicate general characteristics for this ethnic group in multicultural Toronto.
Point 6: and situating the results within the cultural sphere
Response 6: This part of the study was added at the end of the discussion:
Toronto was chosen as a study area because it provides a unique opportunity for research on ethnic minorities – it is the most important immigrant-receiving city in Canada. Toronto attracts about a half of all immigrants who come to Canada. Therefore, when analyzing the quality of urban space by Polish people in Toronto, in addition to the analysis and examples of assessing urban quality in the world, it is worthwhile to present selected studies conducted on different ethnic groups assessing the quality of urban space in the described city.
Carlos Teixeira [130] in his research made an analysis between two ethnic groups living in Toronto, between the Portuguese and the black race. In his research, he presented the strategies and barriers that businesses owned by representatives of ethnic groups face, in order to assess how ancestry affects running a business in the city. The research indicated that Portuguese people differed significantly from black entrepreneurs in that they were more likely to rely on their community (“ethnic”) resources. Black entrepreneurs faced more barriers to starting and/or running their current business, particularly in obtaining loans from financial institutions and banks in Toronto, which contributed to their overall poorer assessment of the quality of urban spaces.
Jennifer Logan & Robert Murdie [131] performed a description of Tibetan refugees settling in Toronto, with a particular focus on whether they were able to obtain affordable homes quickly. The study showed that most Tibetans have had a home in Toronto for less than ten years, and their overall assessment of the quality of urban space is poor.
Meanwhile, Mohammad Quadeer & Sandeep K. Agrawal & Alexander Lovell [132] in their research focused on describing four ethnic group living in Toronto, and trying to assess their satisfaction with living in the city. They chose Chinese immigrants, Jews, Portuguese, and Italians as their research group. Their research found that migrant enclaves are mainly located in suburban areas, where the percentage of homeowners is high, and their satisfaction with the location and quality of life is high.
In his article, Kenneth L. Dion [133] described the Housing New Canadians project, in which he presented the results of a study on immigrants’ perceptions of discrimination in the search for rental housing. In his study, respondents from three immigrant communities – Jamaican, Poles and Somalis – indicated the perceiving housing discrimination they experienced personally and how the discrimination was directed at their migrant group, with Poles rating this aspect much less negatively than the other study groups. Respondents also rated the extent to which their nationality, income level, source of income, immigration status, English language proficiency, ethnicity, religion, and family size contributed to their negative assessment of the discrimination and prejudice they perceived. Jamaican and Somali immigrants perceived more personal and group discrimination than Polish immigrants, who in the survey reported overall satisfaction with their standard of living in Toronto.
Kenneth L. Dion and Kerry Kawakami [134] of the University of Toronto analyzed the quality of life ratings of six ethnic groups, representing minorities (Black, Chinese and Asian) as well as white minorities (Italians, Jews and Portuguese). The study found that Black, Chinese and Asian felt more discrimination against their group than white minorities, especially in economic areas of life regarding finding a job, salary, and social advancement, thus rating urban space worse in these aspects than other study groups.
Drawing on translational theory and previous research on the housing trajectories of new Canadians, Sutama Ghosh [135] analyzed the housing experiences of two “South Asian” groups of in Toronto – Indians from India and people from Bangladesh. Highlighting important differences among the intra-immigrant groups, the study showed how diverse transnational ties influence their choice of neighborhood and housing type, and quality. Indians are dispersed throughout Toronto, while Bangladeshis have gathered in specific parts of the city, often having smaller houses and being less satisfied with life in Toronto than Indians.
Toronto is a major immigrant-receiving city in Canada and contains a wide diversity of ethnic groups. Although Canadians are generally open to immigration, there is evidence that some recent immigrant groups, especially those concentrated in the suburbs of Toronto, do not do well with social integration, which translated into their assessment of the quality of life in the city. This is demonstrated by Murdie R and Ghosh S [136] in their research. Their findings challenge traditional perspectives on ethnic concentration, particularly the spatial assimilation model, and highlight the importance of considering subjective integration, especially life satisfaction in the new country, as a means of mitigating barriers to poor functional integration. In the case of Toronto, spatial concentration does not necessarily imply a lack of integration, although in the case of Asian immigrants who tend to concentrate in suburban enclaves, their withdrawal from urban life is clearly evident.
Kant et al. [137], who published data on the quality of life of aboriginal people in Canada in 2013, are well known. The research team conducted these surveys in the province of Ontario (population 600, 120 households) and British Columbia (population 1500, 275 households). During the study, 316 questionnaires were collected. The questionnaire included questions about satisfaction with: general well-being, education, employment, health, housing, income, and land use. The research showed that the Aboriginal people are not satisfied with their health or quality of life. The research indicate that the development of a national policy based on attention to mental health in Aboriginal people, and assistance from government in this area would significantly increase the quality of life for these peoples in Canada.
Another study on the quality of urban living space was conducted by Wei-Wei and Alicia Garcia [138] on the Chinese diaspora. The study investigated the sociocultural adaptation, and change in settlement quality of life among recent elderly Chinese immigrants in Canada. The study was conducted on 31 elderly men and women wo had recently immigrated from China to Canada, and their main purpose was to explore their perceptions of quality of life in a new place. Their overall perceptions of post-immigration quality of life were marginalized by losses and gains, and depended on multifaceted factors such as language, intergenerational relationship, economic status, and people supporting them.
Point 7: The comparisons are not clear. Why did the author chose Kocaeli for comparison? why Hong Kong (line 509)? (Living conditions determine quality of life almost everywhere) etc.
Response 7: The author believes that in the discussion it is necessary to mention similar research conducted in the world within other metropolises. In addition, describing similar phenomena among other research groups around the world (not only Toronto) aims to make comparisons between different populations around the world, which from the perspective of global cultural and sociological phenomena is very important and interesting.
Point 8: Please refer to the number of new comers to Canada each year 390,000 or a quarter of Million.
Response 8: The sentence in lines 113-114 “It welcomes around a quarter of a million newcomers per year” has been removed. The correct number is 390,000, which is left in line 73.
Reviewer 2 Report
The paper focuses on a widely studied aspect (quality of urban spaces) from a rarely analyzed perspective, as the Polish minority in the GTA is in the centre of it. The author cited numerous relevant and current papers, thus the Introduction section is almost done. However, line 64-88 might be moved to 2.1. subsection, since this content fits better to the spatial scope. Moreover, a solid background about the Polish urbanization processes is suggested to add to the text to prepare the ground for further analyses and point out the importance of analysing the Polish minority from an urban study perspective. The applied methodology is generally well described and explained; there is no need to modify it thoroughly. Nevertheless, please add some information about the general socio-economic status of the analyzed community as a whole to provide basic values regarding the age structure and wellbeing. In the Discussion section (which is a really high-quality one), please add some thoughts about the main limitations of your analysis and potential further assessments. Moreover, please consider providing your assumption whether the general well-being of the respondents how to modify their individual perception regarding the quality of urban space in the area they live. Finally, the last section does not include any summary of the results; please add a short introduction of the study's outcomes to this section. To summarize, the manuscript tried to fill a scientific gap by analysing an important issue from an unusual perspective. The paper includes significant novelty; however, several modifications might be made before acceptance and publication to improve the quality of its methodological basis.
Author Response
Response to Reviewer 2 Comments
Point 1: However, line 64-88 might be moved to 2.1. subsection, since this content fits better to the spatial scope.
Response 2: The reviewer’s proposed paragraph was moved to section 2.1.
Point 2: The applied methodology is generally well described and explained; there is no need to modify it thoroughly. Nevertheless, please add some information about the general socio-economic status of the analyzed community as a whole to provide basic values regarding the age structure and wellbeing.
Response 2: This part of the study was added to section 2.2
To ensure that the sample group was representative, Census Canadian data from 2016 and 2019 were used to calculate the gender and age structure of individuals in each age group relative to the entire Greater Toronto Area Polish population. The most recent census, which includes people of Polish ancestry and is available, is from 2016. It was first used to calculate that there are 523,490 people of Polish ancestry living in Ontario, of which 251,575 are men and 271,910 are women. The next step was to calculate how many people of Polish ancestry live in the studied area. There are 113,300 men and 123,940 women living in the GTA, for a total of 237,245 people. The methodology of the research assumed that the author will examine 0.25% of the population of Polish origin living in the region. After calculations, this gave a total of 593 people, including 283 men and 310 women. For the research to be representative and reliable, it was necessary to calculate the age and gender structure, according to the methodological assumptions. Publicly available censuses do not take into account the current age and gender breakdowns of representatives of the various national minorities in Canada. Therefore, it was assumed, that based on publicly available age and gender breakdown data, the author would perform calculations for Ontario residents to determine what percentage of particular age group is male and what percentage is female. The survey was conducted by age and gender, and 583 properly completed survey questionnaires were used for the final analysis. After establishing the age and gender structure of the Ontario population, the author determined, by calculation, the quantitative structure of the Polish community residing in the GTA.
In order to make the research as representative as possible, seven age groups have been differentiated, with the use of the pre-conceived division. These were the following age groups: 15-19, 20-24, 25-29, 30-39, 40-49, 50-59, over 60 (where women and men were additionally included separately). Conducting research among people aged 15 and over (youth) allowed to be sure that respondents understood the asked questions and answers provided by them were reliable. In this article the research results are presented for all age groups together.
However, the purpose of this article was not to indicate the quality of living standards in particular age groups, but to capture the statistical relationship in the surveyed population of Polish diaspora, and to indicate general characteristics for this ethnic group in multicultural Toronto.
As indicated by the results of the study, the Polish community living in the Greater Toronto Area is characterized by either a high school or college education. The highest percentage of respondents with college education was found in the age range of 30-39 years, where they accounted for 14%, and in the age group over 60 years – 14.7%. In the most numerous group of respondents, over 60 years, 12.6% of respondents have a university degree, and 11% of respondents have high school education. Polish community living in Greater Toronto Area, their overall financial situation is assessed as good – so said 53.3% of respondents, and as very good – 23.2%. It was considered average by 16.5%, and bad by 5.0%. Only three persons in the age group of 15-19 assessed their material situation as very bad, there were no such responses in other age groups. As many as 42.2% in the age group of 30-39 years rated their material assessment as very good and in the age group of 50-59 years were 29.1% such people.
Point 3: In the Discussion section (which is a really high-quality one), please add some thoughts about the main limitations of your analysis and potential further assessments. Moreover, please consider providing your assumption whether the general well-being of the respondents how to modify their individual perception regarding the quality of urban space in the area they live.
Response 3: This part of the study was added at the end of the discussion:
The assumption of the research conducted on the Polish community in the GTA was to construct a model, based on which it was possible to examine the Polish community in many areas of life, concerning the level and quality of life (in such aspects as: cultural, social, economic, educational, legal, geographical, and health). This is a novelty that the author proposes as a research tool to assess the quality and standard of living of the population. The research results presented in this article are only concerned with selected indicators that are used for the general assessment of urban space. The article does not describe the model constructed by the author, nor does it provide a detailed analysis of individual factors, but the constructed model and the results of the research conducted on the Polish community of the GTA provide the possibility for further, reliable, and in-depth analysis of assessing the quality of life in such areas of life as: economic, cultural, social, educational, legal, geographical and health.
Point 4: Finally, the last section does not include any summary of the results; please add a short introduction of the study's outcomes to this section. To summarize, the manuscript tried to fill a scientific gap by analysing an important issue from an unusual perspective.
Response 4: The introduction of the results has been moved from the discussion section – line 563-591 to conclusion.
Reviewer 3 Report
The paper aims at assessing the quality of life by people of Polish ancestry living in the Greater Toronto Area (GTA) with particular emphasis on urban quality, by administering purposely designed questionnaires.
The topic is relevant and the GTA case study could be very interesting.
Nevertheless some weaknesses have been detected in the structure of the article.
The empirical survey collects the perceptions of the respondents, It needs to be integrated by the territorial analysis of the context of living in order to have a better picture of urban services and infrastructures.
It is not clear how the Polish ancestry influences the perception of quality of life respect the other inhabitants with different background.
The discussion needs to be better related to the empirical analysis and the case study methodology.
The research rationale is poorly described and the robustness of the quantitative analysis needs to be better explained.
Reviewer 4 Report
The article is generally written correctly and embedded in the literature on the subject it presents. It also meets the technical requirements imposed by the publishing house. From my point of view, however, its subject matter and center of gravity do not match Land. At the center of this text is a sociological analysis and an assessment of the satisfaction of a certain group of people living in a specific territory. It is not space that is the subject of research here, but the quality of life.
Not this research method. Not this the way of presentation to consider Land publications, too. If the answers of the respondents (geocoded on the maps) were related to what on other maps shows the location of the elements (even generalized ones) about which were asked, I would not hesitate to give a positive opinion on this article (with some fixes). But in my opinion it is not passible to improved text to this form.
In my opinion (this interesting and not devoid of revelatory value) the text should be published, but in such a periodical as Social Sciences or Urban Science, when it comes to the MDPI publishing house). After significant reformulation of the text (towards balancing the quality of life and access to infrastructure), it could be published in Sustainability.
The discussion and conclusion side should also definitely be improved so that it more closely matches the research results.
Round 2
Reviewer 1 Report
no comments
Author Response
The Reviewer indicated no comments. Thanks for all previous comments and suggestions. Everything has been changed according to the previous recommendations of the Reviewer.
Reviewer 2 Report
The author made all the corrections I suggested previously, and the manuscript went through a really thorough revision; consequently, it can be accepted in present form.
Author Response
The Reviewer indicated no remarks or comments. Thanks for all previous comments and suggestions. Everything has been changed according to the previous recommendations of the Reviewer.